# Jasmonic Acid Boosts Physio-Biochemical Activities in *Grewia asiatica* L. under Drought Stress

**DOI:** 10.3390/plants11192480

**Published:** 2022-09-22

**Authors:** Abdul Waheed, Yakupjan Haxim, Gulnaz Kahar, Waqar Islam, Abd Ullah, Khalid Ali Khan, Hamed A. Ghramh, Sajjad Ali, Muhammad Ahsan Asghar, Qinghua Zhao, Daoyuan Zhang

**Affiliations:** 1Xinjiang Key Laboratory of Conservation and Utilization of Plant Gene Resources, Xinjiang Institute of Ecology and Geography, Chinese Academy of Sciences, Urumqi 830011, China; 2Xinjiang Key Laboratory of Desert Plant Roots Ecology and Vegetation Restoration, Xinjiang Institute of Ecology and Geography, Chinese Academy of Sciences, Urumqi 830011, China; 3Research Center for Advanced Materials Science (RCAMS), King Khalid University, Abha 61413, Saudi Arabia; 4Applied College, Mahala Campus, King Khalid University, Abha 61413, Saudi Arabia; 5Unit of Bee Research and Honey Production, Biology Department, Faculty of Science, King Khalid University, Abha 61413, Saudi Arabia; 6Department of Biology, College of Science, King Khalid University, Abha 61413, Saudi Arabia; 7Department of Botany, Bacha Khan University, Charsadda 24461, Pakistan; 8Agricultural Institute, Centre for Agricultural Research, ELKH, 2462 Martonvásár, Hungary; 9Binzhou Vocational College, Binzhou 256603, China

**Keywords:** plant growth, JA application, antioxidant, ascorbate peroxidase, CO_2_ assimilation

## Abstract

It has been shown that jasmonic acid (JA) can alleviate drought stress. Nevertheless, there are still many questions regarding the JA-induced physiological and biochemical mechanisms that underlie the adaptation of plants to drought stress. Hence, the aim of this study was to investigate whether JA application was beneficial for the antioxidant activity, plant performance, and growth of *Grewia asiatica* L. Therefore, a study was conducted on *G. asiatica* plants aged six months, exposing them to 100% and 60% of their field capacity. A JA application was only made when the plants were experiencing moderate drought stress (average stem water potential of 1.0 MPa, considered moderate drought stress), and physiological and biochemical measures were monitored throughout the 14-day period. In contrast to untreated plants, the JA-treated plants displayed an improvement in plant growth by 15.5% and increased CO_2_ assimilation (AN) by 43.9% as well as stomatal conductance (GS) by 42.7% on day 3. The ascorbate peroxidase (APX), glutathione peroxidase (GPX), and superoxide dismutase (SOD) activities of drought-stressed JA-treated plants increased by 87%, 78%, and 60%, respectively, on day 3. In addition, *G. asiatica* plants stressed by drought accumulated 34% more phenolics and 63% more antioxidants when exposed to JA. This study aimed to understand the mechanism by which *G. asiatica* survives in drought conditions by utilizing the JA system.

## 1. Introduction

It is obvious that drought is a significant factor contributing to environmental stress in plants, reducing plant yields by more than 50% [1,2,3]. Plant growth, carbon accumulation, and tissue expansion are negatively affected by drought stress. The studies have demonstrated that drought stress decreases the activity of the ribulose-1,5-biphosphate carboxylase and oxygenase (Rubisco) enzymes and those involved in Calvin and Benson cycle reactions. Additionally, there is low adenosine triphosphate (ATP) production as well as damage to photosystem II (PS II), which results in a decreased photosynthetic rate [4,5]. According to previous studies, stomatal closure is the main cause of photosynthesis inhibition. However, metabolic impairment could also play a role [6,7,8]. Nevertheless, the mechanisms underlying the metabolic impairment under drought stress are not fully understood [9]. The presence of reactive oxygen species (ROS) in chloroplasts, mitochondria, and peroxisomes is a consequence of drought stress in plants. Under drought stress, a number of plant organelles, such as chloroplasts, mitochondria, and peroxisomes, produce reactive oxygen species (ROS). Several studies have shown that the excessive production of ROS damages DNA, proteins, and lipids [10,11].

As a result of drought-induced excess ROS accumulation, various metabolic changes, including the denaturation of protein and membrane degradation, damage to photosynthetic pigments, deoxyribonucleic acid degradation, and the shrinkage of plant growth, occur [12,13]. In response, plants upregulate the biosynthesis of enzymatic and nonenzymatic antioxidants for the detoxification of excess ROS in stressful environments [14,15]. For their ability to catalyze the destruction of reactive oxygen species (ROS), including hydrogen peroxide (H_2_O_2_) and superoxide anions (O^−2^), superoxide dismutase (SOD) and ascorbate peroxidase (APX) provide most of the antioxidant protection [16,17]. In addition to their hydrogen-donating capabilities, nonenzymatic antioxidants, such as phenols, inhibit lipoxygenase and scavenge ROS, leading to drought responses [18,19].

In plants, JA activates systemic acquired resistance (SAR) to work together to protect plants against biotic and abiotic stressors [20,21,22]. JA is involved in plant growth and development and nutrient uptake, which has a direct impact on growth, cell elongation, and the production of photosynthetic pigments as well as source-to-sink regulation under nonstressed conditions. It has been extensively investigated that JA could enhance the drought stress tolerance of plants [23,24,25,26]. A study by Asensio et al. [27] reported that the exogenous application of JA resulted in an improvement in CO_2_ uptake in drought-exposed *Zoysia japonica* plants. In addition, exogenous JA application has also been reported to increase the antioxidant mechanism and reduce the level of lipid peroxidation in drought-stressed maize plants [28]. Therefore, JA application can enhance the growth and development of plants by providing protection against biotic and abiotic stress [29]. However, JA-induced drought stress tolerance in plants is not completely understood at the physiological and biochemical levels [30,31].

Falsa (*G. asiatica*), is a tiny purple fruit that looks like a blueberry and is an important endemic berry that grows worldwide and is very popular in Pakistan. *G. asiatica* is commonly consumed in the summer as a juice or carbonated drink [32]. In addition to phenols, flavonoids, and anthocyanins, blueberries are rich in antioxidants. Globally, Grewia has almost 150 species and is the only genus in the “Tiliaceae” family to produce edible fruit. Several of these species grow in Pakistan. In South Asia, *G. asiatica* is cultivated and grows wild as well, and its fruit is acidic and sour [33]. This species has been subjected to several studies regarding its morphology and physiological characteristics, considering its antioxidant properties [34]. Consequently, we propose that JA application triggers various antioxidant mechanisms, enzymatic and nonenzymatic, that have been implicated in reducing oxidative stress, which benefits photosynthesis and plant growth. We attempted to determine the effects of JA application on the antioxidant enzymatic activities, photosynthetic performance, and plant growth of drought-stressed *G. asiatica*. In this study, we provide new insight into how JA enhances *G. asiatica*’s antioxidant defense system.

## 2. Materials and Methods

### 2.1. Experimentation Conditions and Plant Material

Six-month-old *G. asiatica* plants were obtained using in vitro conditions during this study. In a greenhouse, plants of uniform size were grown in 1.5 L of Andisol soil in plastic pots for two weeks following a randomized complete block design (RCBD). In the greenhouse, the photoperiod was 16/8 h, the temperature was 23 ± 2 °C, the relative humidity was 70%, and the mean photon rate was 300 mol photons/m^2^/s [35]. A two-week acclimation period was followed. The plants were grouped into two groups during the first 10 days of treatment. After 10 days, plants not receiving daily irrigation (NI) attained a stem water potential of about 1 MPa on average, while those that were receiving daily irrigation (DI) were held at 100% field capacity. Both treatments (+JA) were applied simultaneously to NI in response to a single foliar application of 0.5 mM JA; they reached moderate drought stress [36,37]. As a wetting agent, Tween 20 was added to 0.5% *(v/v)* ultrapure water, which was used to dissolve the JA. Ultrapure water containing Tween 20 (−JA) was used as the control solution. The experiment was conducted for 14 days. In vivo leaf samples were collected at different times following JA application (0, 3, 7, and 14 days), and gas exchange levels were measured at each sampling point. They were stored frozen at −80 °C for biochemical analyses.

### 2.2. Water Status and Plant Growth Measurements

#### 2.2.1. Relative Growth Rate (RGR)

Hoffmann and Poorter’s [38] protocol was used to determine the growth rate during the experiment following destructive harvesting. The RGR was determined from the mean natural logarithm-transformed plant dry weights using Equation (1):
RGR = [(lnDW2) − (lnDW1)/(t2 − t1)](1)
where InDW1 and InDW2 are the plant dry weights at times t1 and t2. The time zero corresponded to T1, while the times 3, 7, and 14 corresponded to T2.

#### 2.2.2. Stem Water Potential and Leaf Relative Water Content (LRWC)

Begg and Turner [39] described a Model 1000 Scholander chamber being used to measure the stem water potential (Ψw) on the leaf petiole. The leaves were wrapped in aluminum foil and placed in a plastic bag for 90 min before measurements. For the determination of the LRWC, fresh leaf samples from each replicated plant were harvested, and the fresh weight was determined. Next, the leaf samples were then dipped in double-distilled water at room temperature for 4 h to reach full hydration, and then the turgid weight was measured. Finally, leaf samples were oven-dried for 48 h at 60 °C to calculate the dry weight. The LRWC was determined using the following Equation [40]:RWC = [(fresh weight − dry weight)/(turgid weight − dry weight)] × 100(2)
where FW is the fresh weight, TW is the turgid weight, and DW is the dry weight.

### 2.3. Measurement of Photosynthetic Parameters

A variety of measurements were made to evaluate the photosynthetic efficiency of *G. asiatica* plants, including the electron transport rate (ETR), net CO_2_ assimilation (PN), transpiration (E), the effective quantum yield (ΦPSII), and the stomatal conductance (gs). The in vivo data were collected using an Li-Cor LR6400 cuvette with its light source and the portable CO_2_ analyzer (08:00–10:30 h) as described earlier [41]. The light source (400 mol photons m^−2^/s), temperature (25 °C), humidity, and CO_2_ concentration were maintained by controlling the portable photosynthesis system. Flow rates of 300 mL min^−1^ and a relative humidity of 80% were used to obtain a concentration baseline using external air with CO_2_. Four measurements were taken for each plant. Finally, the water-use efficiency (WUE) was determined as the ratio of PN/E.

### 2.4. Lipid Peroxidation (LP)

We estimated (malondialdehyde) MDA to assess lipid peroxidation. The frozen leaf samples (0.15 g) were homogenized in 3 mL of 5% trichloroacetic acid (TCA) in an ice bath. Upon homogenization, the homogenate was centrifuged for 5 min at 4000 rpm (25 °C). In the next step, 5 mL of 0.5% thiobarbituric acid (TBA) was added to the supernatant. A boiling water bath (100 °C) was used to heat the mixture, and then it was quickly cooled on ice and recentrifuged for 5 min at 1000 rpm [42]. The supernatant was collected and used to measure the absorbance at 450 nm, 532 nm, and 600 nm. Equation (3) was used to calculate MDA equivalents to thiobarbituric acid-reactive substances (TBARS) [43]:
MDA equivalent (nmol/mL) = [(A532 − A600)/157,000] × 106 (3)

### 2.5. An Assessment of Total Phenolic Content and Antioxidant Capacity 

Liquid nitrogen was used for crushing leaf samples, and ethanol (80% *v*/*v*) was used to macerate them. Then, the homogenate was centrifuged for 10 min at 13,000 rpm at 4 °C. The supernatant was collected to determine the amount of phenol and the antioxidant capacity of the supernatant. To determine antioxidant activity, the DPPH (2,2-diphenyl-1-picrylhydrazyl) method was used [44,45], and absorbance was measured at 515 nm. When comparing the antioxidant capacity to fresh weight, mg of Trolox equivalent are expressed per g of fresh weight (mg TE g^−1^ FW). The total phenolic contents of the samples were calculated using the Folin–Ciocalteu method (Singleton and Rossi, 81), while caffeic acid was used as a standard. The total phenol content was calculated using mg of caffeic acid equivalent for each gram of fresh mass (mg CAE g^−1^ FW).

### 2.6. Determination of Glutathione Peroxidase, Superoxide Dismutase, and Ascorbate Peroxidase Activities

Liquid nitrogen was used to grind the leaf samples, and the macerated samples were pretreated in a 50 mM potassium phosphate buffer (K_2_HPO_4_-KH_2_PO_4_, pH 7.0).

The activity of GPX was determined using H_2_O_2_ as a substrate. In order to prepare a reaction mixture, we mixed frozen leaf samples with 100 mM K-P buffer (pH 7.0), 1 mM NaN3, 1 mM EDTA, one unit of GR, 2 mM GSH, and 0.12 mM NADPH. Next, H_2_O_2_ was added to initiate the reaction. Finally, NADPH oxidation was read at 340 nm for 1 min, and the enzymatic activity was determined by the extinction coefficient of 6.62 mM^−1^ cm^−1^ [46].

For SOD and APX determination, the crude extract was centrifuged for 15 min at 11,000× *g*. Nitroblue tetrazolium (NBT) was used as a photochemical indicator of SOD activity [46]. The mixture was diluted with 0.1 mM ethylene diamine tetraacetic acid (EDTA), 13 mM methionine, 20 L of crude extract, and 322 m Nitro blue tetrazolium chloride (NBT) in a solution of potassium phosphate buffer, EDTA (0.1 mM), and NBT (3.2 mM). For the reaction to begin, riboflavin was added. Thereafter, the absorbance of the reaction mixtures was measured at 560 nm after 15 min of illumination. An SOD unit is equivalent to the amount of enzyme inhibiting 50% of the reduction in NBT. The protein content was used as a reference point for enzyme activity. The protein content was determined using the Bradford method [47,48]. The activity of APX (EC. 1.11.1.11) was determined according to the previously described method [49]. The dilution of the crude extract (40 µL) was performed with 1 mL of extraction buffer, 5 µL of H_2_O_2_ (30% *v*/*v*), and 40 µL of ascorbic acid at 10 mM. The molar extinction coefficient of 2.8 mM cm^−1^ was used for the measurement of enzymatic activity.

### 2.7. Analysis of Statistical Data and Experimental Design 

A randomized design was used to conduct three replications of each treatment and time. In order to determine whether the variances were homogeneous and normal, we used the Levene and Kolmogorov–Smirnov tests. A three-way ANOVA was then performed using drought treatments, JA applications, and time post-JA-application as factors. The multiple comparison test was conducted using Tukey’s *p* ≤ 0.05. The analysis was carried out using Sigma Stat v.2.0 (SPSS, Chicago, IL, USA).

## 3. Results

### 3.1. RGR and Plant Water Status in G. asiatica

*G. asiatica* plants experienced moderate drought stress that adversely affected plant growth, which was reduced by 29% compared to well-irrigated plants (control, 100% FC) (Table 1). Despite this, *G. asiatica* plants affected by moderate drought showed faster growth after JA application, with increases of 12.3% RGR, 15.7% RGR, and 14.8% RGR, respectively, when compared to moderate drought-affected plants not treated with JA. In plants with JA application, RGR increased by 12.6% and 16% on days 3 and 7 of the experiment, respectively. RGR was highly significant between plants with and without JA application on day 14 of the experiment. As a result of moderate drought stress, the stem water potential of plants decreased significantly (−0.5 MPa) on day 3 of the experiment in comparison with well-watered plants (Figure 1A). Plants that had been drought-stressed for 7 days responded to JA application by showing similar Ψw to well-watered plants. In contrast, when JA was applied to moderately drought-stressed plants on days 7 and 14, the Ψw reached similar levels as in drought-stressed plants without JA. On the other hand, the well-irrigated plants showed no changes throughout the experiment. Plants that were moderately drought-stressed displayed lower levels of RWC than those that had been well-irrigated (Figure 1B). Furthermore, moderate drought-stressed plants treated with JA had similar RWC values to those that were not treated. Plants with 100% FC with and without exogenous JA showed no substantial difference in RWC (Figure 1B).

### 3.2. The Effect of Drought Stress on Photosynthetic Performance

A significant interaction was observed among irrigation treatments, ΦPSII, CO_2_ assimilation, *gs*, and *E* but not WUE. Drought stress did not affect the levels of ΦPSII and ETR in well-irrigated plants since they remained unchanged (Figure 2). Surprisingly, exogenous JA application increased the ETR (33%) in drought-stressed plants on day 3 of the experiment, while a 41% increment was seen in ΦPSII. The JA-treated and well-irrigated plants also exhibited the same pattern, with increments in the ΦPSII and ETR on days 3 and 7 then a decline on day 14. Moderate drought stress, on the other hand, caused a 30.8% reduction in *AN* in *G. asiatica* plants compared with those exposed to 100% FC at the beginning of the experiment (Figure 3A). Plants in drought conditions with JA treatment displayed a 42.9% increased *AN* on day 3 compared to plants in a moderate drought without JA treatment. Plants that were moderately stressed by drought experienced an increase in AN without JA that reached equivalent levels of *AN* on days 7 and 14. Additionally, exogenous JA increased *AN* by 33.8% in 100% FC plants on day 3 compared with untreated controls; however, plants without JA application exhibited quite similar AN to 100% FC on day 7 and untreated control.

The *gs* of mildly drought-damaged plants was lower than that of well-irrigated plants (Figure 3B). A 41% growth rate increase was observed in moderately stressed plants when exogenous JA was added on day 3 but reduced on days 7 and 14. From day 7 onwards, the threshold for *gs* tended to increase more slowly in well-irrigated JA-treated plants than in plants without the JA treatment. During the experiment, plants that received proper irrigation experienced significantly lower rates (62%) of *E* than those that did not (Figure 3C). The same effect was observed when JA was applied to plants. The plants that were drought-stressed exhibited increasing *E* levels on day 3 compared to plants that were not treated, while plants that were well-irrigated increased their *E* levels on days 7 and 14. Additionally, our study also found that JA application increased WUE on day 3 following moderate drought stress, reaching its maximum value on day 7 of the experiment. However, well-irrigated plants only experienced a small improvement on day 3 (Figure 3D).

### 3.3. Assessment of Antioxidant Activity and Total Phenolic Content

A significant interaction was found between irrigation, the number of days, and the JA treatment in the antioxidant capacity (AC) and the total phenolic content (TPC). A moderately drought-stressed plant’s AC on day 3 was lower when compared to a well-irrigated plant (Figure 4A). As the experiment progressed, the difference between the two groups increased slightly (about 13%). The amount of AC increased by 33% on 7 and 14 when JA was applied to moderately drought-stressed plants. When exogenous JA was applied to well-irrigated plants on day 7, their AC levels were slightly higher, whereas they experienced a drop in AC levels on day 14. The TPC had similar percentages in drought-stressed plants and well-irrigated plants on all days except day 0 (Figure 4B). A remarkable improvement (62%) in TPC levels occurred with exogenous JA application on the 3rd and 7th days in drought-stressed plants. A well-irrigated plant’s TPC value reached its maximum potential on day 3, while a plant treated with JA had not reached its maximum potential on day 3 (Figure 4B).

### 3.4. Effect on LP

The LP of plants of the *G. asiatica* species was determined as a stress indicator. The results showed that *G. asiatica* plants under moderate drought stress conditions had higher lipid peroxidation levels, which were 2.5-fold higher than well-irrigated plants (Figure 4C). Conversely, drought-stressed plants treated with JA had significantly reduced MDA contents starting on day 7, and this lasted until day 14. However, JA-treated plants with well-irrigated soils remained unchanged in their levels of MDA, displaying a mean level of 18 nmol g^−1^ fresh weight during the experiment.

### 3.5. Glutathione Peroxidase Activity, Superoxide Dismutase Activity, and Ascorbate Peroxidase Activity

In order to study the enzymatic antioxidant defense mechanism, the GPX, APX, and SOD activities were measured in *G. asiatica*. The SOD, GPX, and APX activities were significantly influenced by irrigation treatments, time, and JA levels (*p* < 0.05). A significant difference between plants experiencing drought stress and plants with well-irrigated *G. asiatica* was witnessed in the levels of SOD activity during the experiment (Figure 5A). Nevertheless, we detected relatively constant levels of SOD activity in well-irrigated plants. Plants treated with exogenous JA and drought-stressed plants showed significant augmentations in SOD activity. In drought-stressed plants treated with JA, the SOD activity was perceived to be amplified by approximately 85% compared with drought-exposed plants without JA (Figure 5A). Due to drought, the SOD activity in JA-treated plants was diminished, reaching a similar level as in untreated plants on the 14th day. As for the GPX and APX levels, we found that drought-stressed plants displayed two-fold higher activity compared to fully irrigated plants (Figure 5B,C). In moderately drought-stressed *G. asiatica* plants, exogenous JA application significantly enhanced the GPX and APX (60 and 78%) activity on day 3 after JA application. As the experiment progressed, by day 14, the GPX and APX activity contrasted with that of the non-JA-treated plants. A similar pattern was seen in plants that had been well-irrigated, where the GPX and APX activity levels increased gradually from day 3 to day 7, peaking around 32 and 37% higher than in plants that had not been well-watered (Figure 5B,C).

## 4. Discussion

Drought stress is the most acute threat to food security resulting from climate change [50,51,52]. According to Table 1, plants receiving moderate drought stress experienced a 27% decrease in relative growth compared to those receiving well-watered conditions on the 3rd and 7th days. Moderately drought-stressed *G. asiatica* plants displayed similar Ψw and RWC values (Figure 1). During the experiment, plants that were well-watered maintained a Ψw around −0.5 MPa, but plants that had drought stress reached a minimum value of −1.3 MPa when watered at 60% FC. *G. asiatica* plants that were moderately drought-stressed had a reduced RWC level of 6% compared to well-irrigated plants. Several species have been found to suffer negative effects from moderate drought stress, including *Phaseolus vulgaris* [53], *Vaccinium corymbosum* [54], *Malus domestica* [55], and *Punica granatum* [56]. Following 20 days of water restriction, as we reported, severe drought stress resulted in plant growth reduced by five times in *G. asiatica* plants, which was also previously reported in sunflower [57], soybeans [58], and *Zanthoxylum acanthopodium* [59].

As a mechanism to prevent water loss, the plant’s earliest mechanism to cope with drought occurs when its stomata closes as a result of drought stress [60]. The *AN* and *gs* values of plants under drought stress declined by 30.8% and 21.4%, respectively, compared to plants that were well-irrigated (Figure 3). Consequently, the *gs* and *AN* levels were lower in drought-stressed *G. asiatica*, which was responsible for plant growth inhibition. However, metabolic impairment may also affect photosynthesis.

Thus, ROS are reported to be produced in the chloroplasts, mitochondria, and peroxisomes of living cells [61,62,63,64,65]. During the photoreduction of O_2_ and the ground-state oxygen in chloroplasts, ROS are generated via photosystem I (PSI) and PSII, and many types of ROS are produced, such as ^1^O_2_, O_2_^−^, OH^−^, and H_2_O_2_, decreasing photosynthetic performance [66,67]. ROS production causes oxidative stress under drought stress by damaging DNA, proteins, carbohydrates, and lipids [10]. Therefore, we determined oxidative stress using lipid peroxidation in our study, as a previously published study found that lipid peroxidation increased 2.5-fold during moderate drought stress in plants [68,69]. Drought stress affected *G. asiatica* plants biochemically and physiologically, the amount of *gs* increased along with lipid peroxidation, and plant growth declined. Our results indicated that RWC positively correlates with the photosynthetic performance represented by ETR, CO_2_, EQY, SC, and TR, implying that drought stress hinders photosynthetic performance (Figure 6).

Under nonstressed conditions, the growth hormone JA is responsible for controlling plant growth and development [37,38,39]. Current research indicates that JA contributes significantly to reducing the negative effects of drought on plants [70,71]. Plants have been able to tolerate drought stress; however, it is not completely understood how this occurs on the physiological and biochemical levels [72,73]. In drought-stressed *Populus deltoids*, *Conocarpus erectus*, and *Olea europaea* plants, JA improves SOD and APX activities [74,75,76]. It is hypothesized that JA application alleviates oxidative stress and increases photosynthesis in *G. asiatica* plants under drought stress, increasing plant growth while reducing oxidative stress. The *AN* level in drought-stressed plants returned to normal 3 days after JA application. Plants treated with JA grew faster than plants without JA (Figure 3A). Comparing well-watered plants with and without JA, the *AN* of exogenously applied JA increased 3 days after the application. Adding exogenous JA to stressed plants increased the *gs* in a comparable manner to nonstressed plants. The WUE of drought-stressed plants began to improve by day 3 of the experiment and peaked by day 7 (Figure 3D). Furthermore, the plants that were subjected to drought stress exhibited higher CO_2_ uptake and higher water levels in JA-treated plants. As a result, irrigation with JA stimulated the growth of drought-stressed *G. asiatica* plants compared to plants without irrigation under drought stress (Table 1). We found a concordance between our results and the previous findings [77,78], demonstrating an increase in CO_2_ uptake and plant growth by the application of JA to drought-stressed *Zoysia japonica*. In previous studies, drought-stressed *Zea mays* responding to JA treatment grew 25% faster and produced 25% more yield as a result of increased CO_2_ assimilation and stomatal conductance, as reported earlier [79,80,81].

It is also of interest that the intercellular CO_2_ concentration did not change between plants that were treated with JA and those that were untreated. Perhaps there is an advantage to JA in improving metabolism. As can be seen from Figure 3, specifically, on the 3rd day of the experiment, the ETR and ΦPSII were higher due to exogenous JA application. It has also been suggested that JA influenced physiology by increasing Rubisco-xylanase activity, increasing CO_2_ assimilation, and promoting plant growth, all of which take part in and/or fix carbon in chloroplasts [82,83,84]. When *Zea mays* are under drought stress, they show an increase in the activity of Rubisco, and Rubisco activates enzymes as a result of JA application [85,86].

In addition, the authors reported that JA positively influenced the low-level transcription of the Rubisco large subunit (Rbc L), the α-form ribulose-1, 5-bisphosphate carboxylase/oxygenase (ZmRCAα), and the β-form ribulose-1, 5-bisphosphate carboxylase/oxygenase (ZmRCA β). JA is reported to inhibit pathogen invasion into plants under biotic stress by closing stomata [87,88,89]. Stomatal apertures increased significantly after the 3rd day in the moderately drought-stressed *G. asiatica* plants treated with JA (Figure 3B). We obtained similar results to previously reported studies where the researchers found that drought-stressed *Z. mays* and *Hordeum vulgare* plants were shown to open their stomata under JA treatment [90,91,92]. Zamora et al. [93] recently demonstrated that spraying JA did not reduce stomatal apertures in *Arabidopsis thaliana* and *Solanum lycopersicum*. High concentrations of JA or prolonged treatment times are necessary for stomatal closure induced by JA. Therefore, we propose to consider species, JA concentration, and biotic and abiotic stress as influencing factors of JA-induced stomatal closure. Likewise, Ahmad et al. [94] have suggested that in drought conditions JA contributes to increased photosynthesis by maintaining the structure and function of the light-harvesting apparatus through enzymatic and nonenzymatic antioxidants. JA application induced a transient increase in photosynthetic performance on day 3 post-JA-application, as measured by ETR, ΦPSII, *gs*, *AN*, and plant growth. Methylation, glycosylation, and amino acid conjugation may affect the activity and level of JA by metabolizing it into inactive or storage forms. This suggests that, in *G. asiatica*, changes in photosynthetic performance and plant growth can be attributed to JA metabolism modulating its levels and activity. Drought reduces ROS and prevents oxidative stress in plants, as previously discussed by multiple studies [24,25,26]. As a result, *G. asiatica* exhibited enzymatic antioxidant defense mechanisms in the form of SOD and APX. 

The application of exogenous JA significantly enhanced (85%) the SOD activity of drought-stressed plants after the 3rd day of our study compared to drought-stressed plants that did not receive JA (Figure 5A). Despite this, the SOD activity in JA-treated plants surpassed that in the control group on day 14 of the study. The APX activity in drought-stressed *G. asiatica* treated with JA was higher on day 3 (60%) but dwindled slightly on day 14 (similar levels to non-JA-treated plants) (Figure 5B). According to Qiu et al. [95] and Ahmad et al. [96], JA application caused an increase in SOD and APX activities in drought-stressed *Glycine max* and *Phaseolus vulgaris* plants, and we also found the same phenomenon in the current study. Specifically, we propose that the overexpression of the isochorismate synthase gene, which facilitates the biosynthesis of JA and leads to the accumulation of JA, could explain some of our results, which resulted in lower ROS and lipid peroxidation in response to drought in *Hordeum vulgare* [97,98]. Significant increases in TPC (30%) were also observed in plants subjected to drought stress on days 7 and 14 of the study when exogenous JA was applied (Figure 4). Furthermore, JA enhanced plant growth under moderate drought conditions, indicating that its application could save water as well. Citric acid chelates transition metal ions, which serve as ROS scavengers during drought stress because of their high reactivity as hydrogen donors or electron acceptors [99,100]. JA application to drought-stressed *G. asiatica* plants significantly reduced their MDA levels, a measure of lipid peroxidation. Our study showed that both enzyme-dependent and non-enzyme-dependent antioxidant mechanisms can reduce ROS in *G. asiatica* plants exposed to drought stress, which in turn resulted in reduced oxidative stress. As shown in our current results, drought stress is alleviated in *G. asiatica* plants by JA-induced enzymatic and nonenzymatic antioxidant mechanisms, promoting photosynthesis and reducing oxidative stress. Plants of *G. asiatica* under drought stress reacted to JA via enzymatic or nonenzymatic antioxidant mechanisms, but the precise mechanisms remain unclear.

## 5. Conclusions

Our study found that moderate drought stress adversely modulates the physiological and biochemical responses of *G. asiatica* plants. As a result, the exogenous application of JA induced a reduction in MDA levels by stimulating antioxidant activities, which led to enhanced photosynthetic performances, which improved the growth of *G. asiatica* plants under moderate drought conditions. We provide new insights into how *G. asiatica* copes with drought stress through JA-induced physio-biochemical mechanisms. Still, further studies are needed to better understand the molecular mechanisms underlying JA-mediated drought stress tolerance.

## Figures and Tables

**Figure 1 plants-11-02480-f001:**
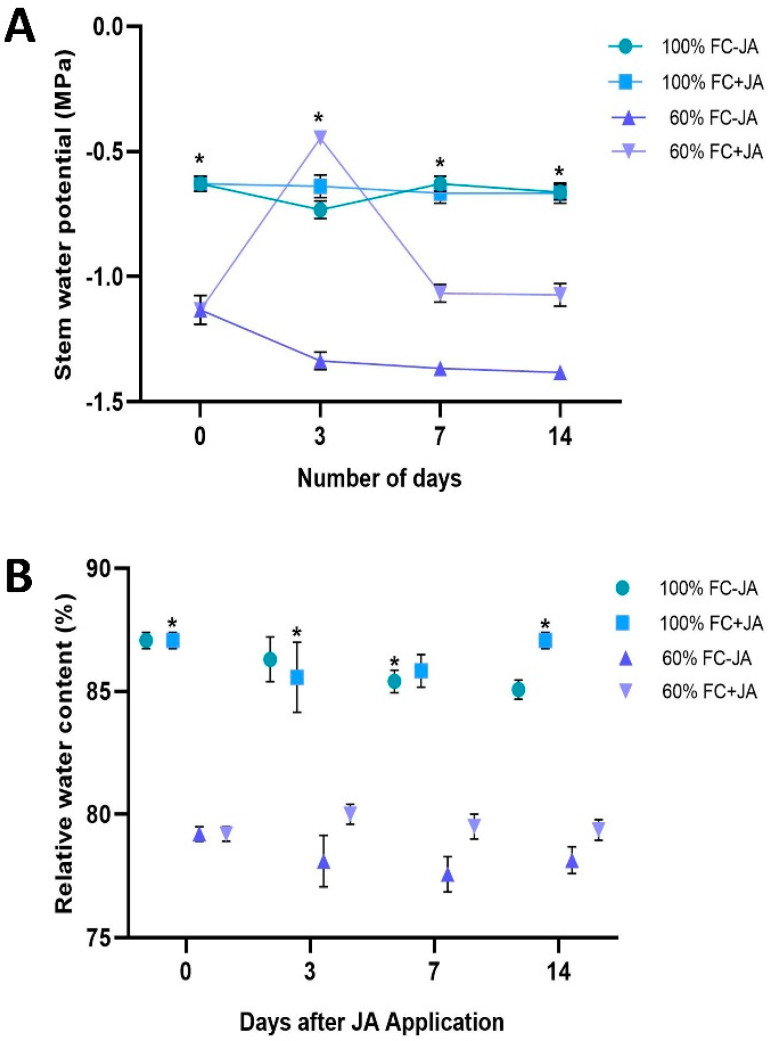
(**A**) Stem water potential (**B**) Relative water content in *G. asiatica* plants grown under 100% and 60% FC and two JA doses (0 and 0.5 mM) at different times post-JA-application. Bars represent the means ± standard errors of the mean (*n* = 3). An asterisk indicates a significant difference between the same JA treatment and the time post-JA-application between irrigation treatments (*p* < 0.05).

**Figure 2 plants-11-02480-f002:**
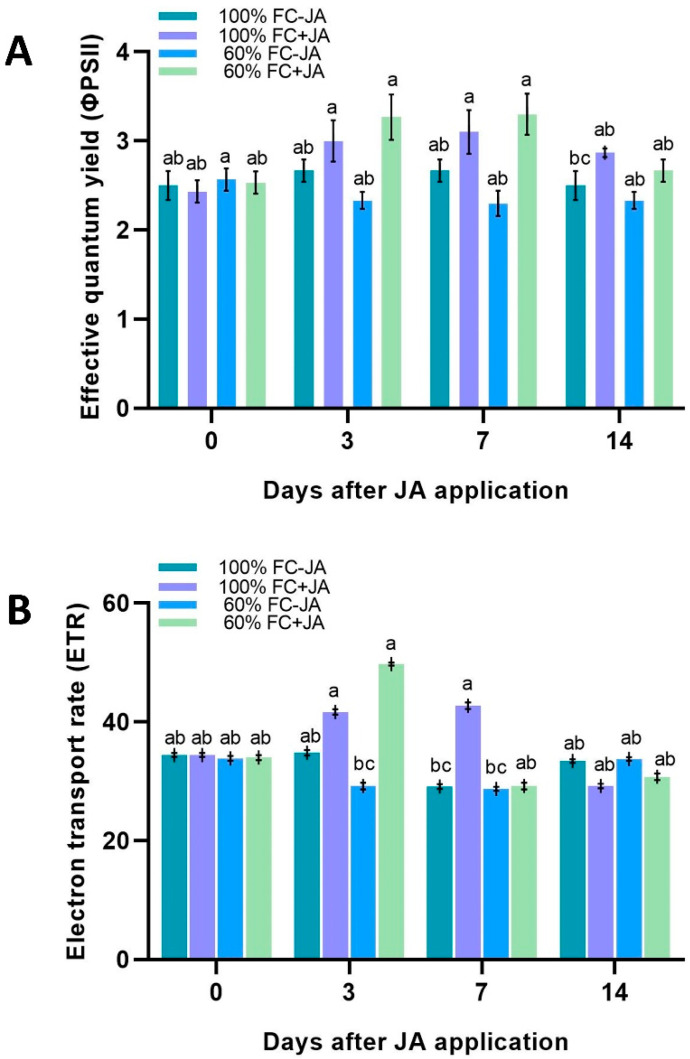
Photochemical parameters in *G. asiatica* L. plants grown under 100% and 60% FC and two JA doses (0 and 0.5 mM) at different times. (**A**) Effective quantum yield (ΦPSII) and (**B**) electron transport rate (ETR). Different letters are indicative of significant differences between irrigation treatments and JA treatments at the time post-JA-application for Tukey’s test (*p* < 0.05).

**Figure 3 plants-11-02480-f003:**
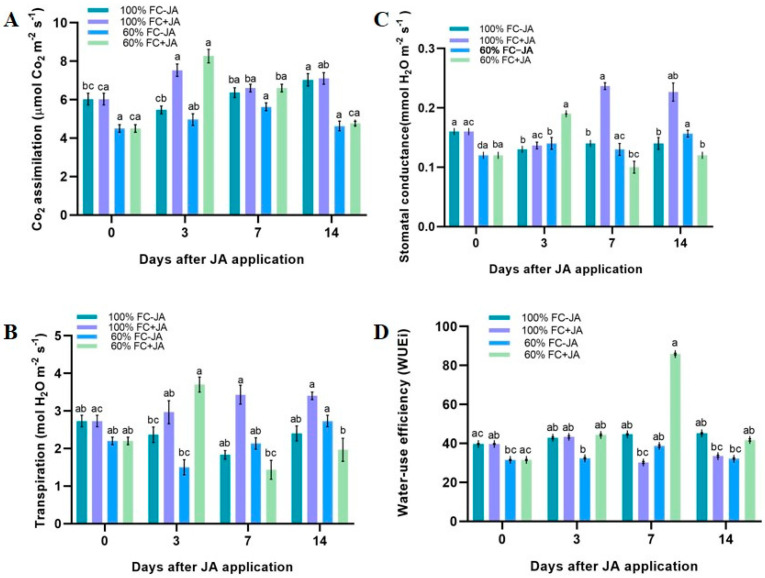
Gas-exchange measurements in *G. asiatica* plants grown under 100% and 60% FC and two JA doses (0 and 0.5 mM) at different times. (**A**) CO_2_ assimilation, (**B**) stomatal conductance, (**C**) transpiration, (**D**) water-use efficiency (WUE). For the same JA treatment and time post-application, different letters indicate significant differences between irrigation treatments according to Tukey’s test (*p* < 0.05).

**Figure 4 plants-11-02480-f004:**
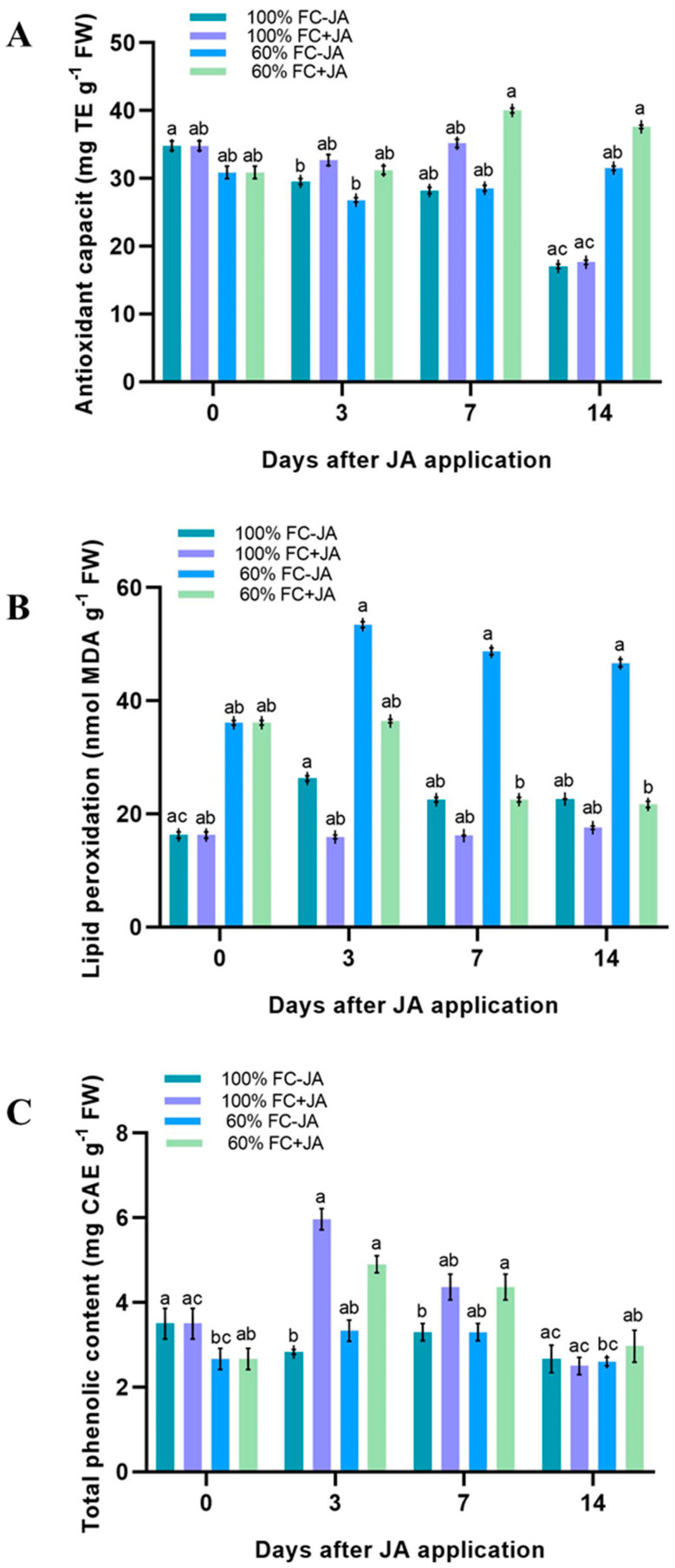
An assessment of the nonenzymatic antioxidant system in *G. asiatica* plants grown under 100% and 60% FC and at two doses of JA (0 and 0.5 mM). (**A**). Total antioxidant capacity, (**B**). phenolic content, and (**C**) lipid peroxidation. As indicated by the Tukey’s test, different letters indicate significant differences between irrigation treatments for the same soil condition and post-application irrigation treatment (*p* < 0.05).

**Figure 5 plants-11-02480-f005:**
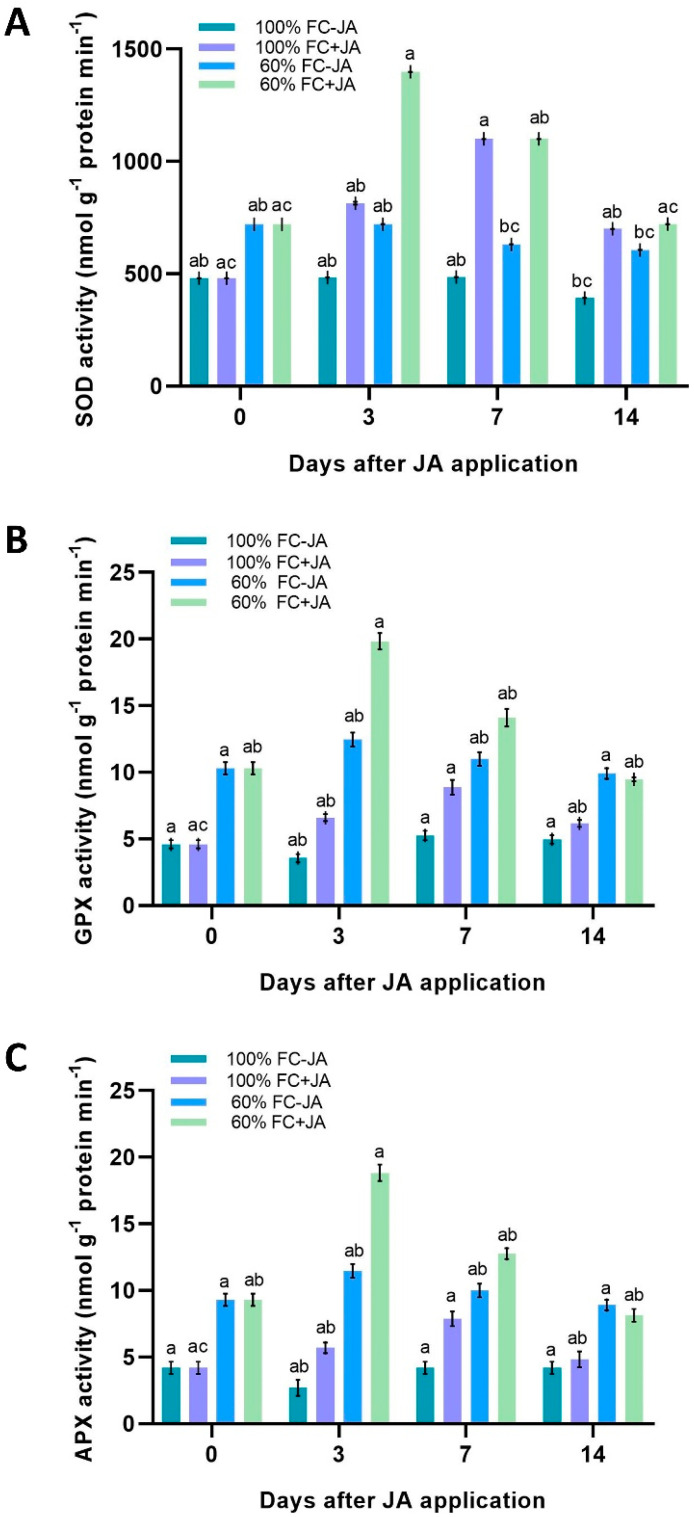
An investigation of the antioxidant system in *G. asiatica* plants grown under 100% FC and 60% FC with two doses (0 and 0.5 mM) of JA at different times after the JA treatment. (**A**). Superoxide dismutase (SOD), (**B**). glutathione peroxidase (GPX) activity, and (**C**) ascorbate peroxidase (APX) activity. Using Tukey’s test, different letters indicate significant differences in irrigation treatments across the experiments, regardless of the JA treatment and time following JA application (*p* < 0.05).

**Figure 6 plants-11-02480-f006:**
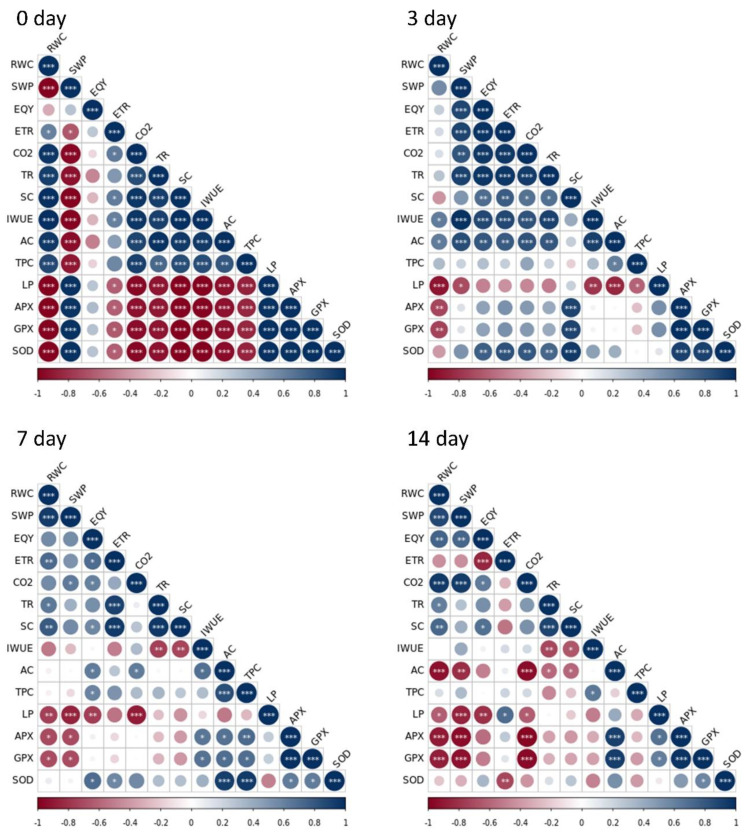
Correlation matrix of morpho-physiological attributes of *G. asiatica* plants under well-watered and water-stressed regimes and exogenous JA application. Relative water content (RWC), stem water potential (SWP), effective quantum yield (ΦPSII, EQY), electron transport rate (ETR), CO_2_ assimilation (CO_2_), transpiration rate (TR), stomatal conductance (SC), intrinsic water-use efficiency (WUE), antioxidant capacity (AC), total phenolic content (TPC), lipid peroxidation (LP), ascorbate peroxidase (APX), glutathione peroxidase activity (GPX), superoxide dismutase (SOD). Pearson’s correlations were calculated and heat maps were visualized by R Statistical Computing Software (version 4.1.3). (*) *p* < 0.05; (**) *p* < 0.01; (***) *p* < 0.001.

**Table 1 plants-11-02480-t001:** A comparison of RGRs of *G. asiatica* plants grown under 100% FC and 60% FC and two different JA doses (0 and 0.5 mM).

Treatments	Day 3	Day 7	Day 14
100% FC − JA	28.12 ± 1.12 Ab	34.66 ± 1.09 Ab	42.51 ± 1.94 Ba
100% FC + JA	58.18 ± 1.17 Aa	72.70 ± 0.99 Aa	79.60 ± 3.11 Aa
60% FC − JA	24.87 ± 1.58 Ab	29.44 ± 1.47 Ab	34.45 ± 1.27 Ab
60% FC + JA	47.70 ± 1.17 Aa	56.31 ± 1.69 Aa	68.53 ± 1.75 Aa

For the same irrigation and JA treatments, Tukey’s test of significance revealed significant differences (*p* < 0.05). Different uppercase letters indicate significant differences among post-JA-application time for the same JA and irrigation treatment according to Tukey’s test (*p* < 0.05). Different lowercase letters indicate significant differences between SA treatment for the same irrigation treatment and day according to Tukey’s test (*p* < 0.05).

## Data Availability

Not applicable.

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
