# Peer review of "Jasmonic Acid Boosts Physio-Biochemical Activities in Grewia asiatica L. under Drought Stress"

_plants, 2022, doi:10.3390/plants11192480_

Round 1
Reviewer 1 Report (Previous Reviewer 3)
The manuscript "Jasmonic acid boosts physio-biochemical activities..." by Waheed et al. is a typical study of the modulation of a plant's stress response, in this case, drought, through the application of jasmonic acid.
The methodology the authors followed makes sense and has been documented well.
The results are now clearly presented, which improves the paper greatly.
Minor points:
- what is Rubicloxylase (line 370) ?
- Table 1 legend: are there differences between A and a ?
Author Response
Wishing you a great day. Thank you for reviewing our manuscript, entitled ‘‘Jasmonic acid boosts physio-biochemical activities in Grewia asiatica L. under drought stress’’. Your comments and suggestion really helped us to improve our manuscript. We tried our best to revise and improve our manuscript based on your comments. Please see the attachment

Reviewer 2 Report (New Reviewer)
*Many punctuation marks such as commas and dots are missing throughout the entire text, for example here *Line 71: plants In addition, or here Line: 96 RCBD) In the
*Line 55: denaturation of protein 85 and…. -What is protein 85??
*Line 87: have been implicated in reducing oxidative stress… -incorrect use of the verb form
*Line 102: 100% field capacity (100%). -Redundant
*Line 108: The frozen leaf samples were then stored at -80 °C for biochemical analyses- They were frozen before stored at –80ºC? or try to say that they were stored frozen at -80ºC
*Line 122: measure stem water potential (Ψw) between 08:00 and 10:00 on the leaf petiole. -I don't understand what they mean
*Line 136: in vivo-on italic
*Line 140: concentration reference- I don’t understand, did you mean a base line
*Line 144: We estimated MDA-I don't know what MDA is, nor can I find where it is explained
*Line 144: lipid per oxidation
*Line 151: TBARS-explain the meaning please
*Line 159: Trolox-explain the meaning please
*Line 169: 1 unit GR- What is GR? please explain
*Line 171: extinction coefficient of 6.62 mM−1 cm−1.-Missing reference
* Line 164: Determination of superoxide dismutase and ascorbate peroxidase activities- missed the Glutathione peroxidase
* Line 175: 322 m NBT- m?
*Line 174-176: 0.1 mM ethylene diamine tetraacetic acid (EDTA), 13 mM methionine, and 20 L of crude extract and 322 m NBT in a solution of potassium phosphate buffer, EDTA (0.1 mM), and NBT (3.2 mM) were added to the mixture.- Poorly constructed sentence with redundancies
* Line 184: was used measurement of enzyme activity- bad construction
* In general, the way in which the results are commented, makes it very difficult for me to follow them. What I would have done and I recommend it to the authors for their future paper, is highlight the important data that means something, and I not comment the most obvious, which only makes it more difficult to follow the content.
*Line 203-204: After 7 days, JA application increased Ψw in drought-stressed plants, reaching levels similar to those of well-irrigated plants. - Looking at the data in Fig 1A, I do not understand how this conclusion can be drawn. please explain it
*Line 223: WUEi- please defined i?
*Line 223: The levels of ΦPSII and ETR were not affected by drought stress in plants, as they remained unchanged in well-irrigated plants (Figure 224 2). - I don't understand this sentence
*Figure 3B, X axis legend is missing
*Figure 3D- Why the water use efficiency is so high only after 7 days with JA and then decrease. Why are the other values measured in this condition not so high?
*Line 262: rose? Slightly
*Line 269: plants (. The
* Line 269: The TPC levels of plants that had been well irrigated reached their maximum potential on day 3, whereas the TPC levels of plants that had not been treated with JA reached their maximum value on day 3 (Figure 4B). - Then what is the difference, I do not understand this sentence. Rephrase
*Line 277: Effect on LP production- I would remove production.
In my opinion, this is one of the most significant data and, on the other hand, little is commented on.
*Line 300: 60and 78%)
*Line 304: peaking? around 32. Rephrase
*Line 357: (Figure. 3D).
*Line 370: Rubicloxylase? Activity
* In the discussion it is noted that the authors are experts on the subject and that they handle a correct and current bibliography and use it well to explain their results.
*Line 420: Figure 6, drought stress is alleviated in G. asiatica plants by JA-induced enzymatic and non-enzymatic antioxidant mechanisms, promote photosynthesis, and reduce oxidative stress. - Could you explain how that conclusion can be drawn from Figure 6? -I'm sorry but Figure 6 I don't understand it. But sorry, to be honest I have not seen a figure like that before either, maybe it is my fault
*in summary: The number of data and graphs is too high for the conclusions that can be drawn from them. Sometimes, less is more. I would completely remake the paper, and I would think which data are the ones that really contribute to something. I encourage the authors to go ahead with the paper but spend a lot more time improving the style and completely redoing it, make it shorter, eliminating unnecessary data and graphs and focusing on those that provide conclusive data.
Author Response
Wishing you a great day. Thank you for reviewing our manuscript, entitled ‘‘Jasmonic acid boosts physio-biochemical activities in Grewia asiatica L. under drought stress’’. Your comments and suggestion really helped us to improve our manuscript. We tried our best to revise and improve our manuscript based on your comments. Please see the attachment

Reviewer 3 Report (New Reviewer)
Title: Jasmonic acid boosts physio-biochemical activities in Grewia asiatica L. under drought stress
General comments: The authors have used Jasmonic acid to evaluate the drought resistance potential for Grewia asiatica L. The study is meeting the scientific standard in terms of technical language. However, the English language of article needs to be revised critically. I suggest major revision for this article. The following suggestions should be considered to improve the manuscript quality before its publication in Plants.
Specific comments:
1- Keywords should be different compared to the words found in title.
2- Re-check the abbreviations which have been mentioned in the whole manuscript. Please elaborate them at least once at its first place in the abstract and other sections of manuscript.
3- Introduction is not supporting your experimental hypothesis.
4- In abstract: rewrite this sentence. “Therefore, this study sought to determine whether JA application had a beneficial effect on antioxidant activity, plant performance, and growth of Grewia asiatica L. in a moderate drought”.
5- Fig. 4: See the titles on Y-axis, two of them starting from capital letter, and one of them starting with small letter. Its unacceptable for quality purpose.
6- Properly mention all figures in the text of the manuscript, at relevant places.
7- In discussion: delete the word “In summary” from the start of the following line; “In summary, drought stress affected G. asiatica plants……..”
8- Although, discussion part is very long, but missing key mechanisms which could be involved in drought resistance by JA-application. Revisit the discussion part. For the purpose, you can have a look on following articles:
(i) https://doi.org/10.3389/fpls.2022.881032
(ii) https://doi.org/10.3389/fpls.2021.799318
9- All scientific names should be italic throughout the manuscript.
10- Please re-write the conclusion. The conclusion is recommended to be supported by the literature studied, put detail of any limitations of this study, describe implications of this study and provide recommendations for future perspectives.
Author Response
Wishing you a great day. Thank you for reviewing our manuscript, entitled ‘‘Jasmonic acid boosts physio-biochemical activities in Grewia asiatica L. under drought stress’’. Your comments and suggestion really helped us to improve our manuscript. We tried our best to revise and improve our manuscript based on your comments. Please see the attachment

Round 2
Reviewer 2 Report (New Reviewer)
Dear author,
I think that, it have been properly answer most of my suggestions.
Thank you
Reviewer 3 Report (New Reviewer)
Although, author have revised the manuscript, but I can still see few flaws in the revised version. My comments mentioning the major flaws in the paper are below. Further, English language is very poor, and nothing is improved. Based on the poor author’s intent to improve the paper quality, I recommend rejection of this manuscript.
1: In first paragraph, at least let the readers know what is drought and what are the causes and why you think its important to explore drought induced consequences?
2: Line 51: However, metabolic impairment could also play a role [6-8]. Play a role in what?
3: Line 53, 54, and 55: “The presence of reactive oxygen species (ROS) in chloroplasts, mitochondria, and peroxisomes is a consequence of drought stress in plants. Under drought stress, a number of plant organelles such as chloroplasts, mitochondria, and peroxisomes produce reactive oxygen species (ROS)”. This is the repetition of same mechanism.
4: The following statement is not scientific. “shrinkage of plant growth”.
5: In introduction: “Therefore, JA application can enhance the growth and development of plants by providing, protection against biotic and abiotic stress”. Do the author think is this sentence is rightly punctuated?
6: Again, a lot of English mistakes in the following: “Falsa/phalsa (G. asiatica), is a tiny looking purple fruit looking like blueberry is an important endemic berry that grows worldwide and very popular berry in Pakistan”.
7: “In a greenhouse, plants of uniform size were grown in 1.5 liters of Andisol soil in plastic pots for two weeks”. What is this? Since when soil is measured in Liters? Your sentence says that 1.5 L Andisol soil was used.
8: How the authors have evaluated that this soil is “Andisol”? Describe the bases of this assumption.
9: “The plants were grouped into two groups during the first 10 days of treatment.” Again, English quality issue.
10: Discussion looks like a repeat of Introduction and explanation of author’s own results. It should describe at least possible mechanisms.
This manuscript is a resubmission of an earlier submission. The following is a list of the peer review reports and author responses from that submission.
Round 1
Reviewer 1 Report
In the article, the influence of jasmonic acid on the physical and biochemical activity of Grewia asiatica L. under dry stress was studied. The work contains a number of new interesting data. The results are presented consistently and well discussed. The work has a small number of minor comments.
Line. 73 - it is necessary to give the full name of the GPX enzyme.
Line 77 - full name must be given. Grewia asiatica.
Line 101- required to lead abbreviation field capacity.
Throughout the text, remove duplication of the full name of the term and its abbreviations, for example, Line 254, 255, 322, 370, etc.
Change places Fig. 4B and 4C.
After correcting technical errors, the article may be published in the journal Plants.
Author Response

(The authors gave the same response as above.)

Reviewer 2 Report
The manuscript by Waheed and co-authors investigates the impact of exogenous Jasmonic acid on photosynthesis and antioxidant system of Grewia asiatica L. plants subjected to moderate drought. The authors have carried out sufficient physiological and biochemical analyzes to determine if JA has a direct effect on the photosynthetic performance and the antioxidant activity of Grewia asiatica L. plants. However, the experimental design is not described correctly, and it is difficult to understand how the experiment was developed and how the JA application was. The text has serious problems with the use of language and has numerous sentences that are very difficult to understand correctly (e.g. sentences in lines 55-57, 69-72, 137-146, 225-227 among others). Material and methods section needs to be improved, most of the methodologies are not well explained and it is not possible to know the procedure (e.g. RGR, RWC, Lipid peroxidation or SOD and APX activities). In addition, It is not explained how the WUEi was calculated. In the manuscript there are various errors in writing and in metric units (e.g. lines 119, 132, 133 or 163 among others). Cited references are not adequate (e.g. lines 46, 61, 76, 98, 366 among others). The authors should clarify what the meaning of the letters in figures 2, 3 and 4. It is necessary to correctly describe the statistical analysis in order to interpret the graphs and results obtained. The article needs many changes to be considered for publication. The scientific sense and the analyzes carried out may be adequate, but in order to be evaluated correctly, a great improvement in the presentation of the manuscript is necessary.
Author Response
The authors extend their gratitude and wish you a great day. Thank you for thoroughly reviewing our manuscript, entitled ‘‘Jasmonic acid boosts physio-biochemical activities in Grewia asiatica L. under drought stress’’. We tried our best to revise and improve our manuscript based on your comments. Your comments and suggestion really helped us to improve our manuscript. Please see the attachment

Reviewer 3 Report
The manuscript "Jasmonic acid boosts physio-biochemical activities..." by Waheed et al. is a typical study of the modulation of a plant's stress response, in this case, drought, through the application of jasmonic acid.
The methodology the authors followed makes sense and has been documented well.
The results are probably good, but a number of problems with the presentation of the data make it hard to assess this:
- Figure 1B should be either a bar plot, or make it a scatterplot with the dots linked as in Fig 1A, Now it seems as if not all samples were taken at the same day (and some even BEFORE application of JA. Also, asterisks are only shown for the 100%FC data and not for the 60% FC data, whereas the first ones do not look significantly different at all and the latter ones (after 3 and 7 days) probably are. See attached picture
- Table 1: what is the meaning of A and a, b...?
- Figures 2, 3, 4, 5: Figure legends talk about asterisks, but the graphs show letters abc....
The manuscript uploaded showed a lot of textual changes (in tracker) which made it hard to read. Still, there are some minor grammatical errors in the text, such as
- line 43 : The studies
- line 60 production of ROS damages
- line 99-10 : The G. asiatica is...
- line 128 held at 100% field capacity (100%)
- line 151-152 Begg and Turner [39] described that scholander chamber... MAKE THAT A scholander chamber was used.... [39].
- line 183 The LP was...
- line 437: When Zea mays plants are....
- line 438: rubisco activatING enzymes (?)
- line 464: after the 3rd day
Other minor comments:
- line 188: what are NEWTONS per gFW (abbreviation nmol MDA...)
- line 435: what is rubicloxylase
- Figure 6: please indicate packages used in R

Round 2
Reviewer 2 Report
Although in the second version of the manuscript there is some improvement with respect to the previous one, there are still numerous errors in the writing and presentation that make it impossible to understand the scientific research carried out. In addition, the authors have not taken into account most of the comments made in the first round of review.
In the material and methods section there are still numerous flaws: The methodology used for the LP assay is not clear, the methodologies used for the WUE or the GPX activity are not described, there are serious scientific writing flaws (lines 175 or 208 among others).
In table 1 there is no explanation about what the statistical letters represent. In the legend of figure 1, the explanation about what the asterisks represent is confusing. In the legends of figures 2, 3 and 4 asterisks are mentioned but in the graphs, statistical differences are represented by letters.
There have also been no changes with respect to the references used. Among others, references 79-81 are misused in 429 or reference number 35 in line 123.
The article still needs a great improvement in terms of writing and presentation so that its scientific quality can be correctly evaluated.